# Entanglement entropy in Lifshitz theories

**Temple He[1]$^\star$, Javier M. Magán[2]$^\flat$ and Stefan Vandoren[3]$^\sharp$**

**1** Center for the Fundamental Laws of Nature, Harvard University,
Cambridge, MA 02138, USA
**2** Instituto Balseiro, Centro Atomico Bariloche, S.C. de Bariloche,
Rio Negro, R8402AGP, Argentina
**3** Institute for Theoretical Physics and Center for Extreme Matter and Emergent Phenomena,
Utrecht University, 3508 TD Utrecht, The Netherlands

$\star$ tmhe@physics.harvard.edu,   $\flat$ javier.magan@cab.cnea.gov.ar,   $\sharp$ s.j.g.vandoren@uu.nl

## Abstract

We discuss and compute entanglement entropy (EE) in (1+1)-dimensional free Lifshitz scalar field theories with arbitrary dynamical exponents. We consider both the subinterval and periodic sublattices in the discretized theory as subsystems. In both cases, we are able to analytically demonstrate that the EE grows linearly as a function of the dynamical exponent. Furthermore, for the subinterval case, we determine that as the dynamical exponent increases, there is a crossover from an area law to a volume law. Lastly, we deform Lifshitz field theories with certain relevant operators and show that the EE decreases from the ultraviolet to the infrared fixed point, giving evidence for a possible $c$-theorem for deformed Lifshitz theories.

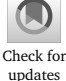

# 1 Introduction

In (1+1)-dimensional relativistic conformal field theories (CFTs), the entanglement entropy (EE) for a line segment at zero temperature obeys a log-law with a coefficient that is proportional to the central charge of the CFT [1, 2]. This result is universal and holds for both weakly and strongly coupled CFTs. It can be reproduced using holography via the celebrated Ryu-Takayanagi (RT) formula [3]. To derive these results, both Lorentz symmetry and scale invariance are used.

In this paper, we break Lorentz symmetry and consider non-relativistic field theories in 1+1 dimensions, but still preserving scale invariance.[1] In particular, we will focus on a class of models known as Lifshitz theories, which have a scaling symmetry under which $x \to \Lambda x$ and $t \to \Lambda^z t$ for $z \in \mathbb{Z}^+$. For $z = 1$, we recover the usual relativistic scaling symmetry. Such $z$ is known as the dynamical exponent of the Lifshitz theory, and such symmetries can arise at quantum critical points in a variety of condensed matter systems. The prototype example is that of a free massless scalar field theory with Hamiltonian[2]

$$H = \frac{1}{2} \int \left[ \pi^2 + \alpha^2 \left( \partial_x^z \phi \right)^2 \right] dx \,, \tag{1.2}$$

where $\pi$ is the conjugate momentum to the scalar field $\phi$, and $\alpha$ has SI units $m^z/s$. For the case where $z = 1$ and $\alpha = c$, we recover the free, massless relativistic field theory. For generic $z$, the model contains particle excitations with dispersion relations $\omega = \alpha k^z$. The scalar field $\phi$ has scaling weight $(z-1)/2$.

Although the theory is Gaussian, the Lifshitz term in (1.2) contains higher spatial derivatives and controls the ultraviolet (UV) behavior of the theory. Therefore, we expect the EE, which is also UV dominated, to depend on $z$, and in fact to grow as a function of $z$. This can be argued by discretizing the model on a one-dimensional lattice. Then, for $z = 1$, there are nearest-neighbor interactions, for $z = 2$ next-to-nearest neighbor interactions, and for larger values of $z$ we get long-range interactions. EE typically grows in the presence of long-range interactions, and hence should increase as $z$ increases. This intuition is confirmed in this paper for the special case of a discretized free $(1+1)$-dimensional real scalar field theory on a circle.

We organize this paper as follows. In Section 2, we study the Lifshitz vacuum EE for a subregion consisting of $N_A$ consecutive lattice sites on a circle with $N$ total lattice sites. By using the recently developed framework of a continuous version of the multi-scale entanglement renormalization ansatz (cMERA) [4], we are able to obtain the universal dependence of EE with respect to the dynamical exponent. In Section 3, we use an approach similar to that

---

[1]We will later also include and discuss mass deformations.

[2]A more general class of Lifshitz models based on free fields is given by the action

$$S = \frac{1}{2} \int \left[ i^m \phi \left( \partial_t^m \phi \right) - \alpha^2 \left( \partial_x^n \phi \right)^2 \right] dx \, dt \,, \tag{1.1}$$

which for $m = 2$ reproduces the Hamiltonian (1.2) with $z = n$. More generally, the Lagrangian (1.1) has a dynamical exponent $z = 2n/m$. However, for $m > 2$, the theory contains higher time derivatives, which implies the quantization of the model, and consequently the study of EE, becomes more difficult and obscure. Thus, we restrict ourselves to the case where $m = 2$, $z = n$, or equivalently the Hamiltonian (1.2), for the remainder of this paper.

studied in [5] to analytically compute the Lifshitz vacuum EE for a subregion consisting of every $p$-th lattice site on the circle. Moreover, in both of these sections, we complement our analytical approach with a numerical one to confirm the results and to explore situations where our analytic approach is not viable. Finally, in Subsection 3.3 we study a renormalization group flow of vacuum EE by deforming the Lifshitz theory in the UV into a relativistic theory in the infrared (IR). We summarize our results in Section 4.

**Note added:** On the day of submission of our paper to the arXiv, the paper [6] appeared. This reference has some overlap with our Section 2. Their study of one-dimensional Lifshitz theories is on an open interval with Dirichlet boundary conditions, whereas we have periodic boundary conditions on a circle. Nevertheless, we see qualitative agreement of the numerical data as a function of the dynamical exponent $z$ and in the massless limit. The authors of [6] also fit their data with a formula, which for $d = 1$ is given, to leading order, by $S^{(z)}(l_A) = \#\left(\frac{l_A}{\epsilon}\right)^{1-\frac{1}{z}} + \cdots$; see equation (3.2) in [6]. While this seems to fit reasonably well in the range of parameters discussed in their paper, namely when $l_A/\epsilon$ is of the same order as $z$, our results suggest this formula fails in the continuum limit, when $z$ is not of the order of $l_A/\epsilon$.

## 2 Subinterval entanglement entropy

In this section, we will first derive a concrete formula for the EE of a free massless Lifshitz scalar field with Hamiltonian (1.2) from cMERA using a straightforward application of the framework introduced in [4]. In the subsequent subsections, we discretize the theory and use numerics to compute the EE. We then compare our numerical results to our result from cMERA.[3]

### 2.1 Scaling arguments from cMERA

Before we start discussing entanglement in Lifshitz theory, let us recall the celebrated result from relativistic CFT that the vacuum EE for a subinterval $A$ of length $l_A$ obeys the area law, which in $1 + 1$ dimensions has a logarithmic dependence given by [1, 2]

$$S = \frac{c}{3} \log\left(\frac{l_A}{\epsilon}\right) + c_0 , \tag{2.3}$$

where $c$ is the central charge of the CFT. Here $\epsilon$ is a short-distance cutoff, and the constant $c_0$ reflects the ambiguity in the cutoff dependence, as multiplying the cutoff by any factor changes the value of the constant $c_0$. The area formula has been reproduced by holography; indeed, since it is universal, holography even obtains the correct result for a free scalar with central charge $c = 1$ [3, 12]. If we discretize the interval with $N_A$ points, so that $l_A = N_A \epsilon$, we would have in the large $N_A$ limit

$$S = \frac{1}{3} \log\left(N_A\right) + c_0 . \tag{2.4}$$

For Lifshitz theories, we no longer know if there is an area law, and whether the coefficient of its "central charge" is universal. However, while we cannot rely on holographic techniques for a free Lifshitz scalar with Hamiltonian (1.2), we can apply a formalism based on cMERA that

---

[3]Due to the lack of the relativistic conformal symmetry for generic $z$, we are unable to carry out the replica trick in the calculation of the EE. The replica method is of course still applicable, but one cannot use the OPE techniques, as in the case of relativistic CFTs. For $z = 2$ and in 2+1 dimensions, there is an analytic approach for computing certain universal and subleading terms in the EE discussed in [7–10], but to our knowledge there is no literature on arbitrary values of $z$ in any dimension. An alternative strategy might be to use the technique developed in [11].

was developed to geometrize EE for free fields in $d$ dimensions [4]. In this approach, we first define the following metric in $d+1$ dimensions:

$$ds^2 = \chi(u)^2 du^2 + \frac{e^{2u}}{\epsilon} d\vec{x}^2 \, , \tag{2.5}$$

where $u = 0, -1, ..., -\infty$ in the discrete MERA and defines the holographic direction in the continuum, $\vec{x}$ the spatial boundary coordinates (in 1+1 dimensions there is only one spatial boundary coordinate, and we will henceforth restrict ourselves to this case), and $\chi(u)$ is defined via

$$\chi(u) = \frac{1}{2} \left( \frac{k \, \partial_k \, \varepsilon(k)}{\varepsilon(k)} \right)_{k=e^u/\epsilon} \, , \tag{2.6}$$

where $\varepsilon(k)$ is the dispersion relation of the free scalar field and $k$ the momentum. Note that the boundary field theory lives at $u = 0$, and the deep interior corresponds to the infrared $u_{IR} = -\infty$.

For our free massless scalar Lifshitz field with Hamiltonian (1.2), we have

$$\varepsilon(k) = \alpha k^z \quad \Rightarrow \quad \chi(u) = \frac{z}{2} \, , \tag{2.7}$$

which means the metric becomes

$$ds^2 = \frac{z^2}{4} du^2 + \frac{e^{2u}}{\epsilon^2} dx^2 \, . \tag{2.8}$$

Note that for $z = 1$, this is just the spatial part of the AdS$_3$ metric. Applying the RT formula by computing the length of the geodesics in the bulk, this metric produces the correct EE formula (2.3). For $z \neq 1$, we apply a trick and notice that we can rewrite the metric (2.8) as

$$d\left(\frac{s}{z}\right)^2 = \frac{1}{4} du^2 + \frac{e^{2u}}{(z\epsilon)^2} dx^2 \, . \tag{2.9}$$

This formula shows something remarkable, namely that the Lifshitz EE for a free massless scalar as obtained from extremizing geodesics in (2.8) is related to that for a relativistic ($z = 1$) free massless scalar by simply rescaling the geodesic length by $z$ and replacing $\epsilon$ by $z\epsilon$. Applying this to (2.4), we obtain the vacuum EE for a free massless Lifshitz scalar[4]

$$S = \frac{z}{3} \log\left(\frac{l_A}{z\epsilon}\right) + z c_0 \, . \tag{2.10}$$

If we choose to discretize the system such that there are $N_A$ points on the interval, each separated by distance $\epsilon$ such that $l_A = N_A \epsilon$, then as long as $N_A \gg z$ so that we are still near the continuum limit, the above equation becomes

$$S = \frac{z}{3} \log\left(\frac{N_A}{z}\right) + z c_0 \, . \tag{2.11}$$

The coefficient $c_0$ is still undetermined and depends on the regularization scheme. Furthermore, notice that for $z \ll N_A$, which is always the case in the continuum, one has an area law (logarithmic in $l_A$), and that the EE is linear in $z$ (as $z \log N_A$ dominates). However, as we will see using numerics in the following subsections, for values of $z$ such that $z \sim N_A$, a crossover happens, and we now have a volume law (linear in $l_A$). This is not so surprising since in the discretized theory, $z$ produces interactions between long-distance neighbors within $z$ lattice sites of each other, so entanglement does not occur only at the boundary. This type of non-local entanglement behavior was specifically studied in [13–15]. In the process of demonstrating this crossover, we will also numerically verify (2.11) to a reasonably good approximation by tuning the coefficient $c_0$.

---

[4]We are assuming the proportionality factor between the geodesic length and the EE is independent of $z$.

## 2.2 Discretization

We begin by considering real $(1+1)$-dimensional free scalar field theory living on a cylinder $S^1 \times \mathbb{R}$, where $\mathbb{R}$ is the time axis and $S^1$ is a spatial circle with circumference $L$. We are particularly interested in the case when our Lifshitz theory has a Hamiltonian of the form

$$H = \frac{1}{2} \int_0^L \left[ \pi^2 + \alpha^2 \left( \partial_x^z \phi \right)^2 + m^2 \phi^2 \right] dx \, , \tag{2.12}$$

where $\phi$ is the scalar field, $\pi$ is its conjugate momentum, $z \in \mathbb{Z}^+$ is the dynamical exponent, and $\alpha^2, m^2 \in \mathbb{R}^+$ are real, positive parameters. In contrast to (1.2), we have introduced a nonzero mass $m$ to avoid divergence issues, which are related to the non-normalizablity of ground states in massless free field theories. The massive extension is certainly interesting in its own right, and by taking the massless limit, we can still study how the EE behaves when the theory becomes scale invariant. Notice that for the case $z = 1$, we recover the usual relativistic free massive scalar theory by setting $\alpha = c$.

In order to bypass the well-known UV divergence in the EE, we discretize the circle into $N$ points, each separated by a distance of $\epsilon \equiv L/N$ that serves as a UV-cutoff. Defining $\phi_j \equiv \sqrt{\frac{m\epsilon}{\hbar}} \phi(j\epsilon)$ and $\pi_j \equiv \sqrt{\frac{\epsilon}{m\hbar}} \pi(j\epsilon)$, so that both $\phi_j$ and $\pi_j$ are dimensionless, we can write the discretized Hamiltonian as

$$H = \frac{m\hbar}{2} \sum_{j=0}^{N-1} \left[ \pi_j^2 + J^{-2} \left( \sum_{r=0}^z \binom{z}{r} (-1)^r \phi_{j+z-r} \right)^2 + \phi_j^2 \right] \, , \qquad J \equiv \frac{m\epsilon^z}{\alpha} \, . \tag{2.13}$$

Note that we impose periodic boundary conditions $\phi_{k+N} \equiv \phi_k$ and $\pi_{k+N} \equiv \pi_k$ for all $k$ since our theory lives on a spatial circle, and that $J$ is a dimensionless coupling constant. Making the assumption that $z < N/2$, which certainly holds in the regime of large $N$, we can rewrite (2.13) as

$$H = \frac{m\hbar}{2} \left( \sum_{j=0}^{N-1} \pi_j^2 + \sum_{i,j=0}^{N-1} \phi_i V_{ij} \phi_j \right) \, , \tag{2.14}$$

where $V_{ij}$ is a symmetric circulant matrix defined as follows:[5]

$$V = \text{circ} \left( J^{-2} \sum_{r=0}^z \binom{z}{r}^2 + 1, -J^{-2} \sum_{r=0}^{z-1} \binom{z}{r}\binom{z}{r+1}, J^{-2} \sum_{r=0}^{z-2} \binom{z}{r}\binom{z}{r+2}, \dots, \right.$$
$$\left. (-1)^z J^{-2}, 0, \dots, 0, (-1)^z J^{-2}, \dots, -J^{-2} \sum_{r=0}^{z-1} \binom{z}{r}\binom{z}{r+1} \right) \, . \tag{2.16}$$

Our goal is to compute the EE for a fixed subinterval $A$, labeled by points $1, \dots, N_A$, as a function of the dynamical exponent $z$. For simplicity, we will assume our system to be in the vacuum state. As was demonstrated in [16–18], it suffices to study the vacuum two-point

---

[5]A circulant matrix is a matrix where every row is a cyclic shift of the row above it, so that it can be defined from the first row alone. As an example,

$$\text{circ}(c_0, c_1, c_2) = \begin{pmatrix} c_0 & c_1 & c_2 \\ c_2 & c_0 & c_1 \\ c_1 & c_2 & c_0 \end{pmatrix} \, . \tag{2.15}$$

functions, which we obtain via mode expansion to be

$$\Phi_{ij} \equiv \langle \phi_i \phi_j \rangle = \frac{1}{2N} \sum_{k=0}^{N-1} \frac{1}{\tilde{\omega}_k} \cos \frac{2\pi(i-j)k}{N} \, ,$$

$$\Pi_{ij} \equiv \langle \pi_i \pi_j \rangle = \frac{1}{2N} \sum_{k=0}^{N-1} \tilde{\omega}_k \cos \frac{2\pi(i-j)k}{N} \, ,$$

(2.17)

where the $\tilde{\omega}_k$'s are the dimensionless eigenvalues of $V$,[6] and the indices $i, j$ run from 1 to $N_A$. The EE is then given by

$$S_A = \sum_{l=0}^{N_A-1} \left[ \left( \lambda_l + \frac{1}{2} \right) \log \left( \lambda_l + \frac{1}{2} \right) - \left( \lambda_l - \frac{1}{2} \right) \log \left( \lambda_l - \frac{1}{2} \right) \right] \, , \qquad (2.18)$$

where $\lambda_l$ are the eigenvalues of the $N_A \times N_A$ matrix $\sqrt{\Phi\Pi}$. Although it is rather difficult to analytically obtain the eigenvalues $\lambda_l$ except for certain very special cases (i.e. when $\Phi\Pi$ is a circulant matrix, as we will explore in the next section), there are no major obstructions to determining the eigenvalues numerically. We simply need to determine the $\tilde{\omega}_k$'s, i.e. the eigenvalues of $V$ in (2.16).

Fortunately, the eigenvalues of circulant matrices are completely known, and we obtain

$$\tilde{\omega}_k^2 = 1 + J^{-2} \sum_{r=0}^{z} \binom{z}{r}^2 + 2J^{-2} \sum_{s=1}^{z} \sum_{r=0}^{z-s} (-1)^s \binom{z}{r} \binom{z}{r+s} \cos \frac{2\pi ks}{N} \, , \qquad (2.19)$$

for $k = 0, \dots, N-1$. Applying Vandermonde's identity

$$\sum_{r=0}^{k} \binom{n}{r} \binom{m}{k-r} = \binom{n+m}{k} \, , \qquad (2.20)$$

we arrive at

$$\tilde{\omega}_k^2 = 1 + \binom{2z}{z} J^{-2} + 2J^{-2} \sum_{s=1}^{z} (-1)^s \binom{2z}{z-s} \cos \frac{2\pi ks}{N} \, . \qquad (2.21)$$

This can be further simplified, after some algebra and using the binomial theorem, to the following simple result:[7]

$$J^2 \tilde{\omega}_k^2 = J^2 + \left( 2 \sin \frac{\pi k}{N} \right)^{2z} \, . \qquad (2.22)$$

We can now use numerics to determine $S_A$ as a function of $z$. Note that because the eigenvalues $\lambda_l$ of $\sqrt{\Phi\Pi}$ are invariant under rescaling of $\tilde{\omega}_k$, we are free to rescale away the $J^2$ on the left-hand-side of (2.22). We are then free to take the massless limit $J \to 0$ without any subtleties.

## 2.3 Numerical results

At large $N$ the dispersion relation (2.22) reads

$$J^2 \tilde{\omega}(x)^2 = J^2 + (2 \sin \pi x)^{2z} \, , \qquad (2.23)$$

---

[6]We obtain the usual dimensionful frequency modes $\omega_k$ (see i.e. [5]) from $\tilde{\omega}_k$ via the relation $\omega_k = m \tilde{\omega}_k$.
[7]We thank Dion Hartmann for pointing this out.

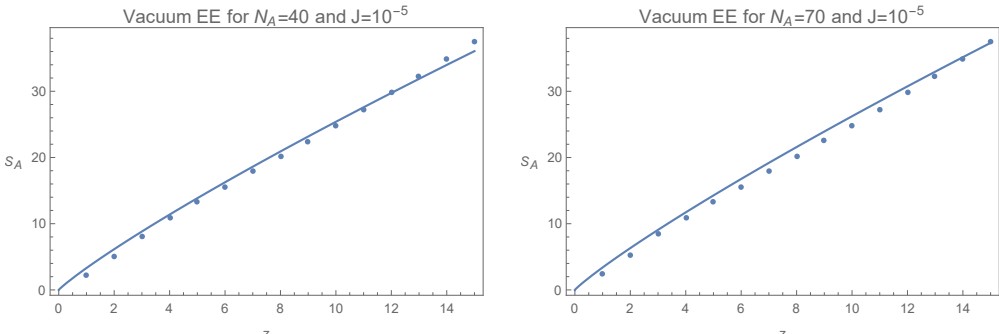

Figure 1: Utilizing the assumption $N \to \infty$, we fixed $N_A = 40, 70$ and $J = 10^{-5}$ and plotted the vacuum EE $S_A$ as a function of $z$. We fitted the data using (2.11) with $c_0 = 1.996$ for the $N_A = 40$ case and $c_0 = 2.080$ for the $N_A = 70$ case, as determined by Mathematica to be the best fit. We expect that the two values of $c_0$ should be the same, and the fit should become exact in the massless continuum limit $J \to 0$ and $N_A \to \infty$.

where $x \equiv k/N$ is a continuous parameter. This is not the continuum limit of the QFT on the circle, because the dispersion in that case would simply be $\omega^2 = m^2 + p^{2z}$, with quantized momenta $p = 2\pi k/L, k \in \mathbb{Z}$. Instead, we keep the cutoff $\epsilon$ small but finite and send $N \to \infty$, which means the circle circumference becomes infinitely long, i.e. $L \to \infty$. Thus, our system is now an infinitely long one-dimensional lattice, precisely the discretized system considered in Section 2.1.

In this limit, the sums in the two-point functions (2.17) become integrals:

$$
\begin{aligned}
\Phi_{ij} &\equiv \langle \phi_i \phi_j \rangle = \frac{1}{2} \int_0^1 \frac{1}{\tilde{\omega}(x)} \cos[2\pi(i-j)x] \, dx \, , \\
\Pi_{ij} &\equiv \langle \pi_i \pi_j \rangle = \frac{1}{2} \int_0^1 \tilde{\omega}(x) \cos[2\pi(i-j)x] \, dx \, ,
\end{aligned}
\tag{2.24}
$$

where $i, j$ run from 1 to $N_A$ for some fixed finite $N_A$. We can then numerically obtain the $N_A$ eigenvalues of $\sqrt{\Phi\Pi}$, and use (2.18) to compute the EE.

After taking the large $N$ limit, the EE $S_A$ is a function of the dimensionless parameters $N_A, z$, and $J$. In Fig. 1, we fixed $N_A$ to be both 40 and 70 and computed $S_A$ as a function of the dynamical exponent $z$ with $J = 10^{-5}$. Although we cannot set $J = 0$ to probe the massless case due to divergence issues in the numerics, $J$ is sufficiently small such that we see that there is qualitative agreement with (2.11). Using Mathematica, we determined that to obtain the best fit, we require $c_0 = 1.996$ given $N_A = 40$ and $c_0 = 2.080$ given $N_A = 70$. Again, we expect these two possible values of $c_0$ to converge to a single number in the massless continuum limit $J \to 0$ and $N_A \to \infty$.

Next, we plotted in Fig. 2 the vacuum EE as a function of $J$, for $N_A = 20$ and different fixed values of $z$. Recalling that small $J$ for fixed $\alpha, m$ corresponds to small lattice spacing (the UV regime) while large $J$ corresponds to large lattice spacing (the IR regime), we see that $S_A$ decreases as we flow from the UV into the IR. This is in accordance with the statement that EE decreases along the renormalization group (RG) flow, a statement proven for relativistic theories in [19], but is not in general true for non-relativistic theories.[8]

Finally, we plotted in Fig. 3 the vacuum EE as a function of $N_A$, as is usually done, for fixed $J$ and different fixed values of $z$. We see that there indeed appears to be a crossover as we go from linear growth (volume law) when $N_A \ll z$ to logarithmic growth (area law)

---

[8]Two such non-relativistic theories whose EEs do not decrease along the RG flow are given in [20].

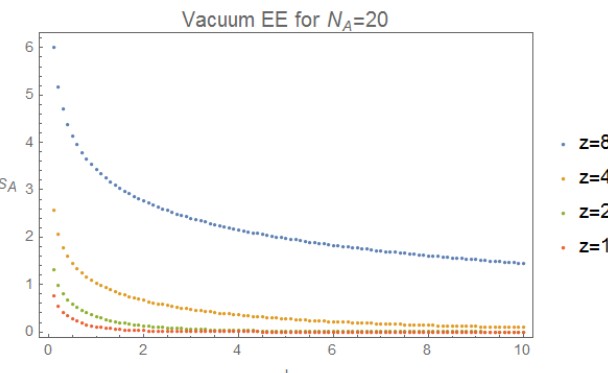

Figure 2: Plot of vacuum EE as a function of $J$ for $N_A = 20$ and different fixed values of $z$. Note that in the UV, the lattice spacing $\epsilon$ is very small, which implies $J \ll 1$, while in the IR, $\epsilon$ is very large, which implies $J \gg 1$. In this $J \gg 1$ regime, the correlation length, inversely proportional to $J$, becomes smaller than the lattice spacing, and we expect the EE to fall off. Thus, we see that regardless of $z$, as we flow from UV to IR, the vacuum EE $S_A$ decreases.

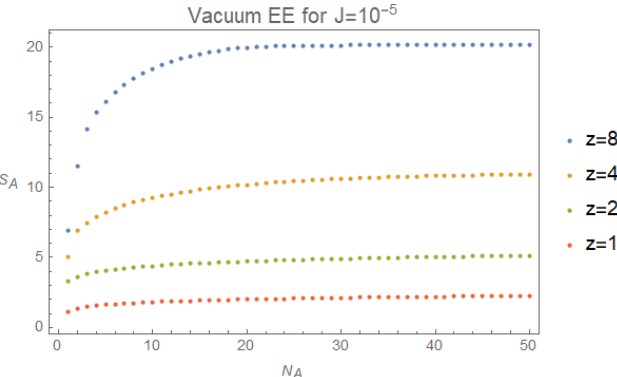

Figure 3: Plot of vacuum EE as a function of the size $N_A$ of the subinterval $A$ for fixed $J = 10^{-5}$ and different fixed values of $z$. For the $z = 1$ case, the plot matches with the Casini-Huerta prediction with central charge $c = 1$. There also appears to be a crossover from logarithmic growth when $N_A > z$ to linear growth when $N_A < z$, as is best visible with the $z = 8$ data points.

when $N_A \gg z$, as predicted by (2.11). For $z = 1$, our numerical result agrees perfectly with the analytic formula obtained by Casini and Huerta in [21, 22], given to be[9]

$$S(l_A) - S(\epsilon) = \frac{1}{3} \log \frac{l_A}{\epsilon} - \frac{1}{2} \log(-\log(m\epsilon)) + \frac{1}{2} \log(-\log(ml_A)) + \mathcal{O}\left(\log^{-2}(ml_A)\right), \quad (2.25)$$

where $l_A > \epsilon$ is the length of subsystem $A$ with $l_A \equiv N_A \epsilon$.[10]

The first term in (2.25) is independent of the mass, and corresponds to the universal area law (which is a log-law in 1+1 dimensions). The other terms diverge in the massless limit and should be subtracted in the conformal limit to avoid divergences. For higher values of $z > 1$, we observe from the data that there appears to be a crossover from an area law, where the

---

[9]A simple sign typo in eq. (96) from [21] was corrected. The formula is for the difference of two EEs and only holds in the $L = N\epsilon \to \infty$ limit, i.e. on the infinite line. Thus, our numerical agreement with (2.25) occurs in the regime when $N \to \infty$, but with the cutoff $\epsilon$ fixed at some small finite value.

[10]Observe further that (2.25) holds when the correlation length, which is inversely proportional to the mass, is much larger than the size of the subsystem, i.e. $ml_A = JN_A \ll 1$, but smaller than the size of the total system, which is obeyed assuming $L \to \infty$. These conditions are satisfied in Figures 1 and 3. For correlation lengths smaller than the size of the subsystem, i.e. $JN_A > 1$, the EE is logarithmic in the correlation length [2], and is relevant only to Figure 2.

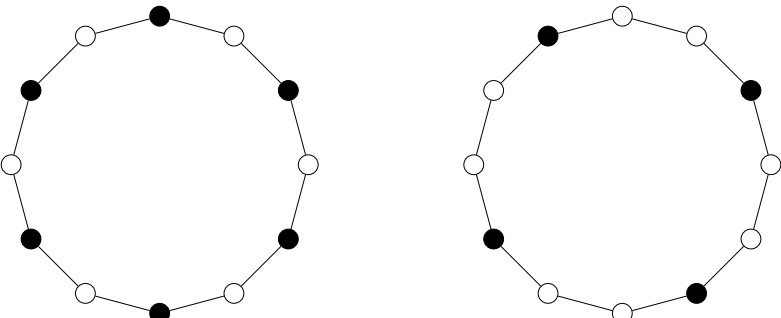

Figure 4: One-dimensional periodic sublattices consisting of $N = 12$ lattice sites, with subsystem $A$ consisting of the $N_A$ filled lattice points such that $N = pN_A$. On the left $N_A = 6$ and $p = 2$, while on the right $N_A = 4$ and $p = 3$. Figure taken from [5].

entropy is logarithimic in $N_A$, when $N_A > z$, to a volume law, where the entropy is linear in $N_A$, when $N_A < z$. As mentioned in the introduction, this makes intuitive sense, since when $z > N_A$, entanglement of subinterval $A$ with the rest of the system is not occurring only at the boundary, but at every lattice site within $A$. Similar nonlocal scenarios were considered in [13–15].

## 3 Sublattice entanglement entropy

### 3.1 Preliminaries

In the previous section, we used cMERA techniques in the massless case, since we were unable to diagonalize the matrix $\sqrt{\Phi\Pi}$ in analytic form. However, if we instead consider a $p$-alternating sublattice, that is, we let our subsystem $A$ to be every $p$-th point as in Fig. 4,[11] then the two-point functions $\Phi$ and $\Pi$ are circulant matrices with the same eigenbasis. In this case, the eigenvalues of $\sqrt{\Phi\Pi}$ can be analytically computed even in the massive scenario.

To write the two-point functions for the $p$-alternating sublattice, we first rescale the indices of the sublattice by $p$ so that instead of labeling the sublattice by the indices $0, p, \dots, (N_A-1)p$, we label it by the indices $0, \dots, N_A-1$. Using the rescaled indices, the two-point functions read

$$
\begin{aligned}
\Phi_{ij} &\equiv \langle \phi_i \phi_j \rangle = \frac{1}{2N} \sum_{k=0}^{N-1} \frac{1}{\tilde{\omega}_k} \cos \frac{2\pi(i-j)k}{N_A} \,, \\
\Pi_{ij} &\equiv \langle \pi_i \pi_j \rangle = \frac{1}{2N} \sum_{k=0}^{N-1} \tilde{\omega}_k \cos \frac{2\pi(i-j)k}{N_A} \,,
\end{aligned}
\tag{3.26}
$$

where $i, j = 0, \dots, N_A - 1$. Except for the fact that the dispersion relation being used here is (2.22), these two-point functions are formally equal to those studied in [5], where it was noticed that the vacuum eigenvalues of $\sqrt{\Phi\Pi}$ are given by

$$
\lambda_l = \frac{1}{2p} \left[ \sum_{i,j=0}^{p-1} \frac{\tilde{\omega}_{l+iN_A}}{\tilde{\omega}_{l+jN_A}} \right]^{1/2} \,, \quad l = 0, 1, \dots, N_A - 1 \,. \tag{3.27}
$$

Substituting this into (2.18) gives us the EE of the subsystem $A$.

We can further simplify our results if we restrict ourselves to the case $N, N_A \gg 1$ with $N/N_A = p$ fixed. Recalling that (2.23) is the dispersion relation in the limit $N \gg 1$, it follows

---

[11]Such systems have for instance also been studied in quantum many-body physics [23–25].

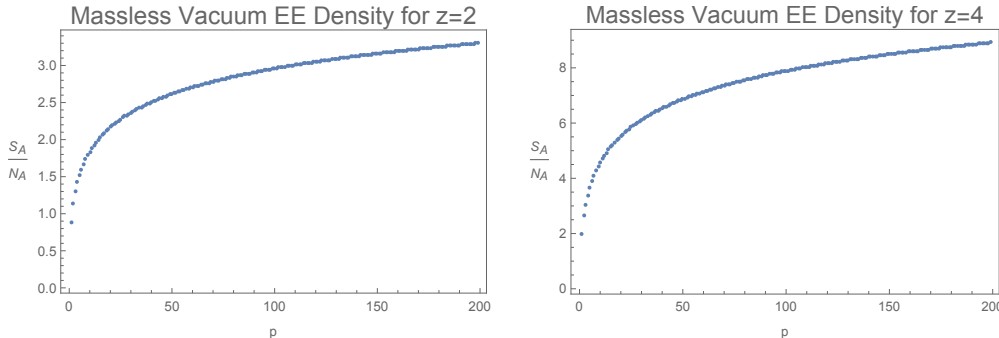

Figure 5: Vacuum EE density of $z = 2$ and $z = 4$ Lifshitz theories as a function of $p$ with $J = 0$, or equivalently, $m = 0$. The vacuum EE density increases without bounds as $p$ increases.

we can also relabel $\lambda_l$ as $\lambda(x)$ and write (3.27) as

$$\lambda(x) = \frac{1}{2p}\left(\sum_{i,j=0}^{p-1} \frac{\tilde{\omega}\left(x + \frac{i}{p}\right)}{\tilde{\omega}\left(x + \frac{j}{p}\right)}\right)^{1/2}, \tag{3.28}$$

where $x \equiv l/N$ becomes a continuous variable in the range $x \in [0, 1/p]$. This in turn implies that we can replace the sum in (2.18) by an integral, and we determine the EE density to be

$$\frac{S_A}{N_A} = p \int_0^{1/p} \left[\left(\lambda(x) + \frac{1}{2}\right)\log\left(\lambda(x) + \frac{1}{2}\right) - \left(\lambda(x) - \frac{1}{2}\right)\log\left(\lambda(x) - \frac{1}{2}\right)\right] dx. \tag{3.29}$$

We remark that there is a caveat here if $m \to 0$, which implies $J \to 0$. In that case, the zero mode $\lambda(0)$ diverges as $J^{-1/2}$, which implies $S_A/N_A$ has a divergent zero mode of the form $\log J/N_A$. This divergence in the massless limit was also found in [5] for $z = 1$. However, as long as we take $N_A \to \infty$ sufficiently fast such that this term vanishes, there will be no divergence in $S_A/N_A$ even for $J \to 0$, and we can ignore such issues.[12] We will henceforth assume this is the case. Modulo this subtlety, it is then clear from (3.29) that the EE is linear in $N_A$ and follows a volume law, i.e. the EE is extensive.

## 3.2 Results

Let us concentrate on the massless case, where the dispersion relation (2.22) reduces to

$$J^2 \tilde{\omega}(x)^2 = (2\sin \pi x)^{2z}, \quad \text{where} \quad J \to 0, \tag{3.30}$$

assuming we take $N_A \to \infty$ fast enough so that $\log J/N_A \to 0$. As we mentioned already under (2.22), the prefactor $J^2$ on the left-hand-side is harmless when we take $J \to 0$ since it cancels out once we compute $\lambda(x)$. Substituting this into (3.28) and (3.29), we obtain the vacuum EE density as a function of $p$ and $z$. The results for $z = 2$ and $z = 4$ are plotted in Fig. 5 as a function of $p$.

One may notice that $S_A/N_A$ in each plot increases as $p$ increases, and wonder whether there is an upper bound on the EE density. We will now show that no such bound exists, and that the vacuum EE density diverges as $p \to \infty$. To see this, note that in that limit, we can approximate the sums in (3.28) as integrals to obtain

$$\lambda(x) = \frac{1}{2}\sqrt{\int_0^1 \frac{1}{\tilde{\omega}(x + y)} dy \int_0^1 \tilde{\omega}(x + y) dy}. \tag{3.31}$$

---

[12]Note that in Section 2, we set $N_A = 20$, which is why the zero mode divergence is present when we take $m \to 0$.

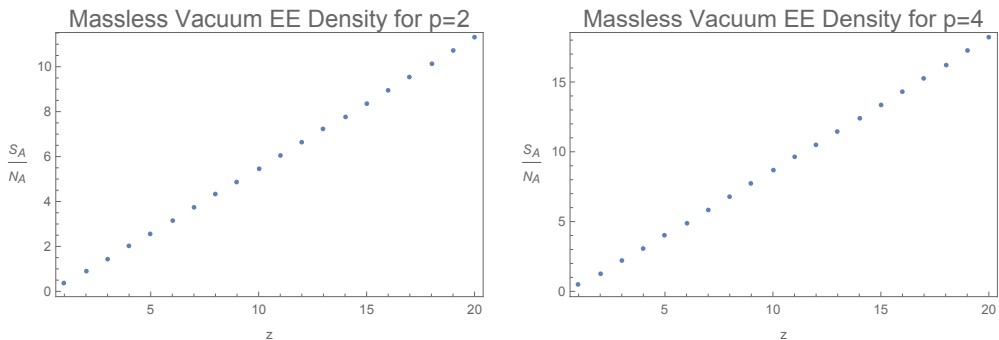

Figure 6: Vacuum EE density for $p = 2$ and $p = 4$ as a function of $z$. The vacuum EE density becomes a linear function of $z$ for sufficiently large $z$.

Observing the fact that $\tilde{\omega}$ is periodic with period 1, it is straightforward to show that the integrals are independent of $x$. It follows that $\lambda(x)$ is in fact independent of $x$, which means by (3.29) that the EE is extensive and is simply given by

$$\frac{S_A}{N_A} = \left(\lambda(0) + \frac{1}{2}\right) \log\left(\lambda(0) + \frac{1}{2}\right) - \left(\lambda(0) - \frac{1}{2}\right) \log\left(\lambda(0) - \frac{1}{2}\right) . \tag{3.32}$$

To compute $\lambda(0)$, we substitute (3.30) into (3.31). Using the fact that for any $z \geq 1$,

$$\int_0^1 \frac{1}{\sin^z \pi x}\, dx \to \infty , \tag{3.33}$$

it immediately follows that $\lambda(0) \to \infty$, which in turn implies $S_A/N_A \to \infty$, thus proving that the vacuum EE density for a massless theory diverges as $p \to \infty$.

Alternatively, we can also fix $p$ and plot the vacuum EE density as a function of $z$. For the cases $p = 2$ and $p = 4$, we have plotted this in Fig. 6. Note that in both cases, the vacuum EE density appears to be linear in $z$. We shall prove here for the simple $p = 2$ case that this linear behavior always holds in the regime $z \gg 1$; the proof for the case of an arbitrary $p$ is given in Appendix A. Using (3.28), we compute

$$\lambda(x) = \frac{1}{4}\left(\cot^{z/2} \pi x + \tan^{z/2} \pi x\right) . \tag{3.34}$$

Before substituting this expression into (3.29) to obtain the vacuum EE density, note that in the large $z$ limit, $\lambda(x)$ diverges as a function of $z$ for all $x \in [0, 1/2)$ except for the point $x = 1/4$. As $\lambda(x) \gg 1$ in the large $z$ limit is violated only on a set of measure zero, we can make the approximation $\lambda(x) \gg 1$ within the integral and approximate (3.29) as

$$\frac{S_A}{N_A} = 2 \int_0^{1/2} \log \lambda(x)\, dx + \mathcal{O}\left(z^0\right) . \tag{3.35}$$

It follows in the large $z$ limit,

$$\begin{aligned}
\frac{S_A}{N_A} &= 4 \int_0^{1/4} \left(\frac{z}{2} \log \cot \pi x + \log\left(1 + \tan^z \pi x\right)\right) dx + \mathcal{O}\left(z^0\right) \\
&= 2z \int_0^{1/4} \log \cot \pi x\, dx + \mathcal{O}\left(z^0\right) \\
&= \frac{2G_C}{\pi} z + \mathcal{O}\left(z^0\right) ,
\end{aligned} \tag{3.36}$$

where $G_C \approx 0.916$ is Catalan's constant. This implies that for sufficiently large $z$, the vacuum EE density for $p = 2$ is linear in $z$ with slope $2G_C/\pi$. This result is coherent with the result of the previous section, which basically states that, up to cutoff non-universal ambiguities, the introduction of a dynamical exponent in the theory renormalizes the central charge $c \to zc$.

### 3.3 Renormalization group flow

We now turn our attention to studying how the sublattice EE changes as we perturb the Lifshitz theory (2.12) with relevant operators. As an example, we choose $\mathscr{O} = (\partial_x \phi)^2$ as a relevant operator that makes the theory flow to a relativistic theory in the IR:

$$H = \frac{1}{2} \int_0^L \left[ \pi^2 + c^2 (\partial_x \phi)^2 + \alpha^2 \left( \partial_x^z \phi \right)^2 + m^2 \phi^2 \right] \mathrm{d}x , \qquad (3.37)$$

where $c$ is the speed of light. More generally, we can perturb the Hamiltonian (2.12) with a relevant operator $\mathscr{O} = (\partial_x^{z_{IR}} \phi)^2$ to generate a flow from a Lifshitz model with dynamical exponent $z$ in the UV to another Lifshitz model with dynamical exponent $z_{IR} < z$ in the IR,[13] provided we set the mass to zero. As the massless limit is obtained straightforwardly from taking $m \to 0$, we will keep the mass term and discuss the massless limit as a special case below.

Discretizing the theory as before, our Hamiltonian becomes

$$H = \frac{m\hbar}{2} \sum_{j=0}^{N-1} \left[ \pi_j^2 + \tilde{J}^{-2} (\phi_{j+1} - \phi_j)^2 + J^{-2} \left( \sum_{r=0}^{z} \binom{z}{r} (-1)^r \phi_{j+z-r} \right)^2 + \phi_j^2 \right] , \qquad (3.38)$$

where $\tilde{J} \equiv \frac{m\epsilon}{c}$ and $J \equiv \frac{m\epsilon^z}{\alpha}$ are two dimensionless parameters. We can rewrite this Hamiltonian in the form (2.14), with $V$ given by

$$V = \mathrm{circ} \left( 2\tilde{J}^{-2} + J^{-2} \sum_{r=0}^{z} \binom{z}{r}^2 + 1, -\tilde{J}^{-2} - J^{-2} \sum_{r=0}^{z-1} \binom{z}{r} \binom{z}{r+1}, \right.$$
$$J^{-2} \sum_{r=0}^{z-2} \binom{z}{r} \binom{z}{r+2}, \dots, (-1)^z J^{-2}, 0, \dots, \qquad (3.39)$$
$$\left. 0, (-1)^z J^{-2}, \dots, -\tilde{J}^{-2} - J^{-2} \sum_{r=0}^{z-1} \binom{z}{r} \binom{z}{r+1} \right) .$$

For reasons that will soon be clear, let us denote the dimensionless eigenvalues of $V$ as $\tilde{\omega}(l/N)$ instead of $\tilde{\omega}_l$, where $l = 0, \dots, N-1$. Then we have

$$\tilde{J}^2 \tilde{\omega} \left( \frac{l}{N} \right)^2 = \tilde{J}^2 + 4 \sin^2 \frac{\pi l}{N} + \frac{\tilde{J}^2}{J^2} \left( 2 \sin \frac{\pi l}{N} \right)^{2z} . \qquad (3.40)$$

Restricting ourselves again to the case $N, N_A \gg 1$ with $N/N_A = p$ fixed, so that we can approximate $x \equiv l/N$ as a continous parameter, we can write our dispersion relation as

$$\tilde{J}^2 \tilde{\omega}(x)^2 = \tilde{J}^2 + 4 \sin^2 \pi x + \frac{\tilde{J}^2}{J^2} (2 \sin \pi x)^{2z} . \qquad (3.41)$$

---

[13]This can reversed if we had instead put the Lifshitz anisotropy on the time derivatives. An example of this is the Lagrangian (1.1) with $n = 1$, so that $z = 2/m$. In this case, we flow from larger $m$ to smaller $m$, which corresponds to flowing from a Lifshitz model with dynamical exponent $z$ in the UV to one with dynamical exponent $z_{IR} > z$ in the IR. We will not study such theories and leave them for future research.

Let us now study how this dispersion relation changes under a particular choice of real-space renormalization. The blocking procedure that we will employ groups $p$ points together at each step, i.e. we have $N \to N/p$ and $N_A \to N_A/p$.[14] We can do this iteratively, so that after $k$ blockings, we have

$$N_k \equiv \frac{N}{p^k}, \quad \epsilon_k \equiv \frac{L}{N_k} = \frac{L}{N} p^k, \quad \tilde{J}_k \equiv \frac{m\epsilon_k}{c} = p^k \tilde{J}_0, \quad J_k \equiv \frac{m\epsilon_k^z}{\alpha} = p^{zk} J_0 , \tag{3.42}$$

where $\tilde{J}_0$ and $J_0$ are the values of the coupling constants before blocking. Thus, after $k$ blockings, the dispersion relation (3.41) becomes

$$\tilde{J}_k^2 \tilde{\omega}_k(x)^2 = \tilde{J}_0^2 p^{2k} + 4\sin^2 \pi x + \frac{\tilde{J}_0^2/J_0^2}{p^{(2z-2)k}} (2\sin \pi x)^{2z} . \tag{3.43}$$

Note that the subscript $k$ is now used to denote the number of blockings performed, which is why we used $\tilde{\omega}(l/N)$ in (3.40) instead to denote the eigenvalues of $V$. Substituting (3.43) into (3.28) and (3.29), we can now compute the vacuum EE density for this deformed Lifshitz theory after each blocking step.

To gain intuition, we plot the vacuum EE density for $z = 2, 4$ and $p = 2, 10$. We picked very small $\tilde{J}_0$ as for large $N \equiv L/\epsilon_0$, $\tilde{J}_0 \equiv m\epsilon_0/c \ll 1$; on the contrary, we expect for $z > 1$ that $\tilde{J}_0/J_0 = \alpha/\left(c\,\epsilon_0^{z-1}\right) \gg 1$. The results are shown in Fig. 7.

First, consider the left side of each of the plots, which is when $k = 0$ and we are deep in the UV regime. Noting that $\tilde{J}_0 \ll 1$ and $\tilde{J}_0/J_0 \gg 1$ in (3.41), our dispersion relation becomes with $k = 0$

$$J_0^2 \tilde{\omega}_0(x)^2 = (2\sin \pi x)^{2z} . \tag{3.44}$$

This is just the dispersion relation (3.30) for a massless pure Lifshitz theory, which we already studied in the previous subsection; the relevant plots for the vacuum EE density as a function of $p$ and $z$ are respectively given in Figs. 5 and 6.

In the opposite regime, consider the limit in which $k \to \infty$ and we are deep in the IR regime. If $\tilde{J}_0 \neq 0$, or equivalently $m \neq 0$, then as $k$ increases, $\tilde{J}_k$ increases while $\tilde{J}_k/J_k$ decreases. It follows from (3.43) that the dispersion relation asymptotes to

$$\tilde{\omega}_\infty(x)^2 = 1 . \tag{3.45}$$

It follows immediately via (3.28) and (3.29) that the vacuum EE density vanishes. This makes sense, since in the IR regime, both $\tilde{J}$ and $J$ become large, which means the term $\phi_j^2$ in (3.38) dominates. This term does not couple different oscillators, and thus there cannot be any entanglement.

However, for the massless case, when $\tilde{J}_0 = 0$, the EE density $S_A/N_A$ asymptotes to a non-trivial value when $k$ becomes large. Recalling that $\tilde{J}_k/J_k$ decreases to zero as we flow into the IR, all dependence on $z$ drops out, and up to a divergent prefactor $\tilde{J}_\infty$ that drops out when computing the EE, the dispersion relation (3.43) reduces to that for a massless relativistic theory:

$$\tilde{J}_\infty^2 \tilde{\omega}_\infty(x)^2 = 4\sin^2 \pi x . \tag{3.46}$$

Thus, as expected, this deformed massless Lifshitz theory flows from a pure Lifshitz theory in the UV to a relativistic theory in the IR. Substituting this into (3.28) and (3.29), we obtain the

---

[14]For simplicity, we decided to choose the blocking parameter $p$ to be the same as that of the $p$-sublattice. This choice is by no means unique, and we could have instead grouped $n \neq p$ points together at each step. In that case, the results for the RG flow would then depend on $n$ and $p$ separately.

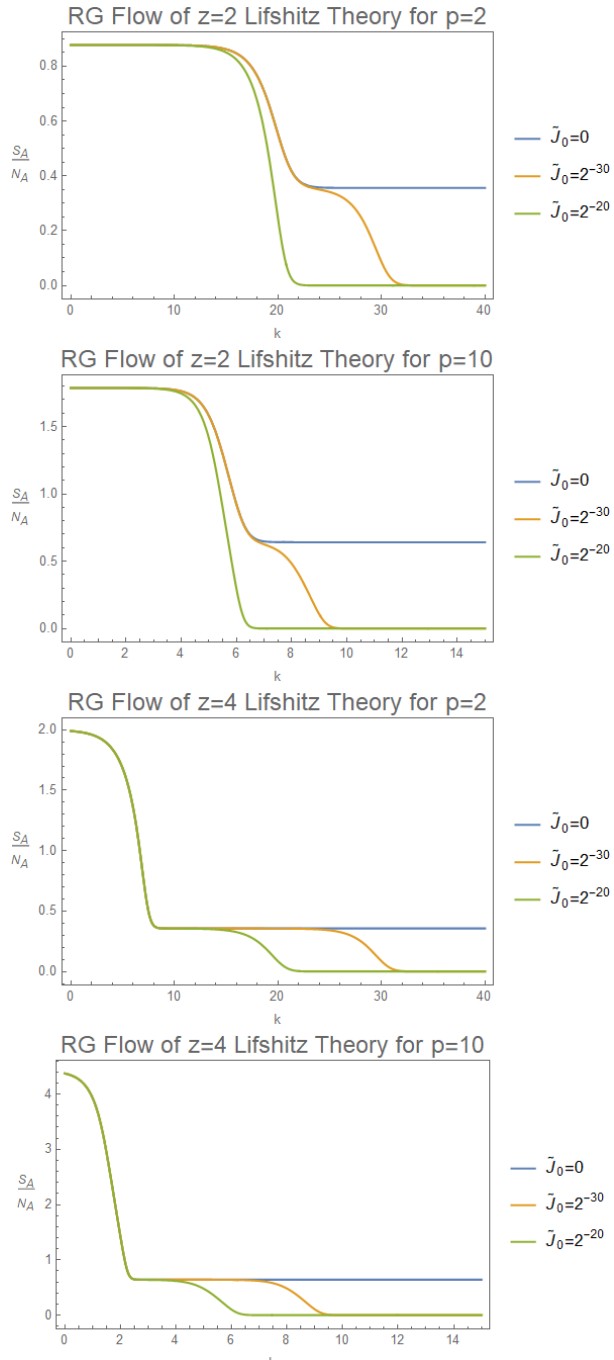

**Figure 7:** Plots of the RG flow of the vacuum EE density for Lifshitz theories in the continuum limit $N \to \infty$ with $z = 2, 4$ and $p = 2, 10$. The first row is for $z = 2$, while the second row is for $z = 4$. We examine the cases when $\tilde{J}_0 = 0, 2^{-30}$, and $2^{-20}$ while fixing the ratio $\tilde{J}_0/J_0 = 2^{20}$ (for the $\tilde{J}_0 = 0$ case we fix the ratio by taking the appropriate limit). The UV regime corresponds to small $k$, and as $k$ increases, we flow into the IR regime.

asymptotic vacuum EE density in the IR for the $\tilde{J}_0 = 0$ curves in Fig. 7. We plot this vacuum EE density in the IR as a function of $p$ in Fig. 8. Note that just as in the pure massless Lifshitz theories analyzed in the previous subsection, the vacuum EE density diverges as $p \to \infty$. This is apparent if we substitute the IR dispersion relation (3.46) into (3.31), and noting the

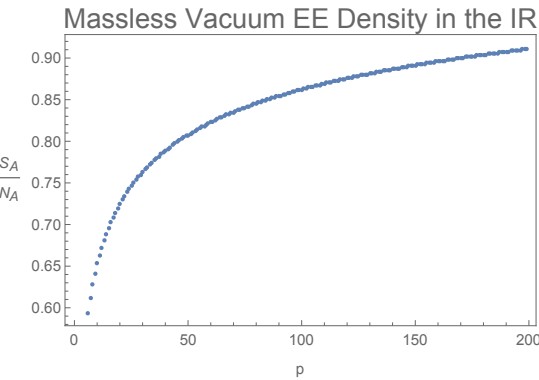

Figure 8: Asymptotic vacuum EE density of a deformed massless Lifshitz theory as a function of $p$. This is in the IR regime, in which the theory reduces to the massless $z = 1$ relativistic theory.

divergence (3.33) holds for $z = 1$.[15]

It is clear from the plots in Fig. 7 that the vacuum EE density of the $p$-alternating lattices is monotonically decreasing along the RG flow. Indeed, we will prove that for any fixed $z \geq 1$ and $p \geq 1$, the vacuum EE density in the UV will always be greater than or equal to that in the IR. This is the weak version of a $c$-theorem, and although the plots give evidence that the strong version of a $c$-theorem also holds, i.e. the vacuum EE density is monotonically decreasing for any fixed $z \geq 1$ and $p \geq 1$, we will not prove the strong version here.

To proceed with the proof of the weak $c$-theorem, we note that it is obviously satisfied for the massive case, since the vacuum EE density in the IR vanishes. In this case, the dispersion relation in the UV is given by (3.44), and that in the IR is given by (3.46), or equivalently obtained from (3.44) by setting $z = 1$. It follows that if we can prove that $S_A/N_A$ increases as $z$ increases, then we are done. For the case $p = 1$, there is nothing to prove since subsystem $A$ is the whole system and hence the EE vanishes along the entire RG flow. Thus, we can restrict ourselves to $p \geq 2$. In particular, this means we can use our intermediate result in Appendix A, which is that $\lambda(x)$ grows for fixed $x \in [0, 1/p]$ and $p \geq 2$ as $z$ increases, i.e. see (A.49). To complete our proof, we only need to show that the vacuum EE density defined in (3.29) grows as a function of $\lambda$. Direct computation yields

$$\frac{\partial}{\partial \lambda(x)} \left( \frac{S_A}{N_A} \right) = p \int_0^{1/p} \log \left( \frac{\lambda(x) + \frac{1}{2}}{\lambda(x) - \frac{1}{2}} \right) dx \ . \tag{3.47}$$

This is positive since the integrand is positive, thus completing the proof of a weak $c$-theorem for these deformed Lifshitz scalar field theories in 1+1 dimensions.

# 4 Conclusions

Entanglement entropy (EE) is nowadays a subject of active research in many areas of theoretical physics. In high energy theory, the EEs of holographic CFTs in particular have been under intense scrutiny, as the RT formula provides a way to straightforwardly compute the EE of a spatial region in the boundary CFT. On the other hand, there has been relatively fewer studies on the EEs of holographic Lifshitz theories.

While the focus of our paper was not on holographic Lifshitz theories, we were driven by such motivations to study entanglement in the simplest Lifshitz theories – a Gaussian theory

---

[15]Alternatively, because the deformed Lifshitz theory in the IR is simply a relativistic theory, one may use the results of Subsection 4.3.1 in [5], where this divergence is explicitly computed.

in $1 + 1$ dimensions. In particular, we wanted to understand specifically the dependence of the EE on the dynamical exponent $z$. In Section 2, we studied the usual subinterval EE by using the recently developed cMERA techniques. These techniques allowed us to obtain the universal scaling of EE with the dynamical exponent in the massless case. The result, given by formula (2.11), states that the EE of a Lifshitz theory with dynamical exponent $z$ is just the EE of a relativistic CFT with an effective central charge rescaled by the dynamical exponent $c \rightarrow zc$.

In Section 3, we studied the EE associated to a $p$-alternating sublattice. The advantage of studying this type of subsystem, as noted in [5], is that the diagonalization of the product of correlation matrices and the entanglement spectrum can be obtained analytically even in the massive case, since both correlation matrices are circulant and share the same eigenvectors. In this context, we were able to analytically show that for large $z$, the vacuum EE again grows linearly with $z$, in agreement with the results obtained from the subinterval entanglement section.

Finally, having analytical control allows us to consider RG flows, in which we flow from a Lifshitz theory in the UV to a relativistic CFT (or to a decoupled theory in which the mass term dominates) in the IR. We were able to prove that the vacuum EE density in the UV is always greater than or equal to that in the IR, thus demonstrating the existence of a weak $c$-theorem for these non-relativistic, Lifshitz deformed theories as well. The numerical plots also appear to indicate the presence of a strong $c$-theorem, for which the vacuum EE density decreases monotonically along the RG flow from the UV to the IR, but we do not have an analytic proof for this at the moment. This Lifshitz $c$-theorems could perhaps be generically proven if one could show also for the interacting cases that introducing a dynamical exponent just rescales the central charge. In such case, Lifshitz $c$-theorems would directly follow from the relativistic CFT ones, but as of now this is an open question.

# Acknowledgements

It is a pleasure to thank Dion Hartmann, Marina Huerta, and Gonzalo Torroba for interesting discussions. This work was supported in part by the Netherlands Organisation for Scientific Research (NWO) under the VICI grant 680-47-603, the Delta-Institute for Theoretical Physics (D-ITP) that is funded by the Dutch Ministry of Education, Culture and Science (OCW), and by the Simons foundation through the It From Qubit Simons collaboration. This work was also supported in part by DOE grant DE-FG02-91ER40654. TH would like to thank Perimeter Institute for its hospitality, during which part of this work was completed. Research at Perimeter Institute is supported by the Government of Canada through the Department of Innovation, Science and Economic Development and by the Province of Ontario through the Ministry of Research, Innovation and Science.

# A  Linearity of vacuum EE density

We prove in this appendix that in the limit of large $z$, the vacuum EE density as a function of $z$ is linear, thereby generalizing the linear behavior seen in Fig. 6 to arbitrary $p \geq 2$. An intermediate result we will show along the way is that the the eigenvalues $\lambda(x)$ grow as a function of $z$ almost everywhere in $x$,[16] which is used in Subsection 3.3 to prove a weak

---

[16] The one exception to this is when $p = 2$ and $x = 1/4$, in which the eigenvalues $\lambda(x)$ remain constant as a function of $z$ by (3.34). However, this occurs on a set of measure zero and therefore won't affect the analysis, as was discussed below (3.34). We will henceforth ignore this subtlety in the appendix.

$c$-theorem.

Our starting point is the dispersion relation in the deep UV given by (3.44). This is the result for a pure massless Lifshitz theory. Substituting it into (3.28), we obtain

$$
\lambda_z(x) = \frac{1}{2p}\left( \sum_{i,j=0}^{p-1} \frac{\sin^z\left[ \pi\left( x + \frac{i}{p}\right)\right]}{\sin^z\left[ \pi\left( x + \frac{j}{p}\right)\right]} \right)^{1/2} , \tag{A.48}
$$

where $x \in [0, 1/p)$, and we denoted the eigenvalue by a subscript $z$ to indicate its dependence on $z$. We now claim that $\lambda_z(x)$ diverges as a function of $z$, for all values of $x \in [0, 1/p)$. To prove this, it suffices to show

$$
\frac{\partial \lambda_z^2}{\partial z} > 0 , \quad \frac{\partial^2 \lambda_z^2}{\partial z^2} \geq 0 , \tag{A.49}
$$

as this will prove that $\lambda_z^2$ (and hence $\lambda_z$ as well) has a positive slope and is a convex function of $z$. Denoting

$$
f_i(x) \equiv \sin\left[ \pi\left( x + \frac{i}{p}\right)\right] , \tag{A.50}
$$

which are positive on the interval $x \in [0, 1/p)$, we compute

$$
\frac{\partial \lambda_z(x)^2}{\partial z} = \frac{1}{4p^2} \sum_{i>j}^{p-1} \left( \frac{f_i(x)^z}{f_j(x)^z} - \frac{f_j(x)^z}{f_i(x)^z}\right) \log \frac{f_i(x)}{f_j(x)} . \tag{A.51}
$$

We note that every term in this sum is nonnegative, and equals zero only if $f_i(x) = f_j(x)$.[17] In particular, since $f_1(x) > f_0(x)$ for all $x \in [0, 1/p)$ for $p \geq 2$ (ignoring the subtlety in footnote 15), the first term in the sum is in fact strictly positive for all $x \in [0, 1/p)$. It follows the right-hand-side is strictly positive, proving $\partial \lambda_z^2/\partial z > 0$. This result will be used to prove the claims in Subsection 3.3.

Next, to show that $\lambda_z(x)$ is convex as a function of $z$, we differentiate (A.51) with respect to $z$ to obtain

$$
\frac{\partial^2 \lambda_z(x)^2}{\partial z^2} = \frac{1}{4p^2} \sum_{i>j}^{p-1} \left( \frac{f_i(x)^z}{f_j(x)^z} + \frac{f_j(x)^z}{f_i(x)^z}\right) \left( \log \frac{f_i(x)}{f_j(x)}\right)^2 . \tag{A.52}
$$

It is obvious that every term in the sum is nonnegative. This proves the second inequality in (A.49), and thus proving that for $p \geq 2$ and $x \in [0, 1/p)$, $\lambda_z(x)$ diverges in the limit of large $z$.

Because $\lambda_z(x)$ diverges in the limit of large $z$ almost everywhere, this means we may approximate (3.29) in that limit as

$$
\begin{aligned}
\frac{S_A}{N_A} &= p \int_0^{1/p} \log \lambda_z(x)\, \mathrm{d}x + \mathcal{O}\left( z^0\right) \\
&= p \int_0^{1/p} \frac{1}{2}\left[ \log\left( \sum_{i=0}^{p-1} \sin^z\left[ \pi\left( x + \frac{i}{p}\right)\right]\right) + \log\left( \sum_{j=0}^{p-1} \sin^{-z}\left[ \pi\left( x + \frac{j}{p}\right)\right]\right)\right] \mathrm{d}x \\
&\qquad\qquad + \mathcal{O}\left( z^0\right) \\
&= p \int_0^{1/p} \frac{1}{2}\left[ \log\left( \sum_{i=0}^{p-1} f_i(x)^z\right) + \log\left( \sum_{j=0}^{p-1} f_j(x)^{-z}\right)\right] \mathrm{d}x + \mathcal{O}\left( z^0\right) .
\end{aligned} \tag{A.53}
$$

---

[17]If we formally set $z = 0$, then every term in the sum is zero. However, we restrict ourselves to $z \geq 1$.

As we mentioned above, each term in the sums inside the bracket is nonegative. Let us first focus on the first sum. For any fixed $x$, we denote $f_M(x)$ as the maximum among the $f_i(x)$'s; note that the index $M$ implicitly can depend on $x$. We then can write the sum as

$$\log\left(\sum_{i=0}^{p-1} f_i(x)^z\right) = \log f_M(x)^z + \log\left(1 + \sum_{i \neq M}^{p-1} \frac{f_i(x)^z}{f_M(x)^z}\right). \tag{A.54}$$

In the limit of large $z$, the second term on the right-hand-side either vanishes if $f_i(x) < f_M(x)$ for $i \neq M$, or it contributes a term of order $\mathcal{O}(z^0)$ if $f_i(x) = f_M(x)$. In either case, we can ignore it to leading order in $z$, thus leaving us with $\log f_M(x)^z$. Likewise, we could have just as easily considered the second sum in (A.53). Letting $f_m(x)$ be the minimum among the $f_i(x)$'s, in the large $z$ limit, the second sum to leading order becomes $\log f_m(x)^{-z}$. It follows we only need to determine what are the minimum and maximum among the $f_i(x)$'s for a given $x$. We proceed by considering two cases separately.

First, consider the case when $p$ is odd. We want to determine for each $x \in [0, 1/p)$, which term in the sums of (A.53) dominates. Now, $\sin \pi x$ is the largest in the interval $x \in \left[\frac{p-1}{2p}, \frac{p+1}{2p}\right)$. Thus, the first term in (A.53) can be approximated as

$$p \int_0^{1/p} \frac{1}{2} \log\left(\sin^z\left[\pi\left(x + \frac{p-1}{2p}\right)\right]\right) dx + \mathcal{O}(z^0). \tag{A.55}$$

On the other hand, the integration interval in which $\sin \pi x$ is the smallest is $x \in \left[0, \frac{1}{2p}\right) \cup \left[1 - \frac{1}{2p}, 1\right)$. Thus, we can approximate the second term in (A.53) to be

$$p\left[\int_0^{1/2p} \frac{1}{2} \log\left(\sin^{-z} \pi x\right) dx + \int_{1/2p}^{1/p} \frac{1}{2} \log\left(\sin^{-z}\left[\pi\left(x + \frac{p-1}{p}\right)\right]\right) dx\right] + \mathcal{O}(z^0). \tag{A.56}$$

Substituting these approximations back into (A.53), we obtain

$$\frac{S_A}{N_A} = \frac{zp}{2}\left[\int_0^{1/p} \log\left(\sin\left[\pi\left(x + \frac{p-1}{2p}\right)\right]\right) dx - \int_0^{1/2p} \log\left(\sin \pi x\right) dx \right.$$
$$\left. - \int_{1/2p}^{1/p} \log\left(\sin\left[\pi\left(x + \frac{p-1}{p}\right)\right]\right) dx\right] + \mathcal{O}(z^0), \tag{A.57}$$

which means $S_A/N_A$ is linear in $z$. Although we've only considered above for the case when $p$ is odd, the analysis works out in a similar fashion for the case when $p$ is even, and the final expression is in fact the same as (A.57). This completes the proof that for any fixed $p$, the vacuum EE density is linear in $z$ in the regime of large $z$.

As a final check of (A.57), let us show that it reduces to (3.36) for the case $p = 2$. In this special case, we obtain using (A.57)

$$\frac{S_A}{N_A} = z \int_0^{1/4} \log \frac{1 + \cot \pi x}{1 - \tan \pi x} dx + \mathcal{O}(z^0)$$
$$= z \int_0^{1/4} \left(\log \cot \pi x + \log\left(\frac{\cot \pi x + 1}{\cot \pi x - 1}\right)\right) dx + \mathcal{O}(z^0). \tag{A.58}$$

This reduces to (3.36) since

$$\int_0^{1/4} \log\left(\frac{\cot \pi x + 1}{\cot \pi x - 1}\right) dx = \int_0^{1/4} \log\left(\frac{\cot\left(\frac{\pi}{4} - \pi x\right) + 1}{\cot\left(\frac{\pi}{4} - \pi x\right) - 1}\right) dx = \int_0^{1/4} \log \cot \pi x \, dx. \tag{A.59}$$

This completes the proof of our claims.

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
