# Peer review of "Entanglement Entropy in Lifshitz Theories"

_SciPost Physics, doi:SciPost Phys. 3, 034 (2017)_

## Round 2 · Referee Report · Anonymous (Referee 1) · 2017-6-19

Strengths

1-Studies entanglement entropy in a new class of 1+1d theories.
2-Relatively clean presentation, writing and organization

Weaknesses

1-Few substantial analytic results
2-Numerical results are not interpreted carefully.
3-No broad or even unexpected conclusions.

Report

This study explores the entanglement entropy (EE) of Lifshitz theories in 1+1d for arbitrary dynamical exponent $z>1$. This is a previously unexplored area, and one that might be fruitful to understand, e.g. how standard rules of entanglement such as the "area law" may be modified in various situations, or how entanglement should behave in holographic Lifshitz theories. Unfortunately, the authors do not pursue the subject far enough to reach interesting conclusions.

The paper consists of two main parts: the first studies EE on a traditional bipartite subinterval, and the second studies it on a periodic entanglement cut. The first section is entirely numerical whereas the second includes some analytic components. The paper concludes that in Lifshitz theories EE may or may not obey the area law, EE increases linearly with z in some cut geometries, and EE decreases along RG flows in these theories.

There are several crucial areas where the work should be improved. It is unknown how the EE depends on the parameters of the theory --- the dynamic exponent and the mass --- and it would be interesting to learn how they qualitatively change the scaling of the entanglement. In other words, one should try to answer the question: what are the general principles for entanglement with higher $z$? By and large, this work performs numerics that demonstrate the behavior of the entanglement, but does not go on to understand the "physical behavior" responsible. Moreover, the analytical content of the work is relegated to a non-standard and less-useful $p$-periodic entanglement cut.

In conclusion, this work is on an interesting topic and its results are correct, but does not go far enough in providing insight and intuition on its subject matter.

Requested changes

1- Equation (1.2) has a typo. The entire paragraph could be condensed to a sentence, since (1.2) is not needed in the sequel except for in a footnote.

2- Page 3 states "Due to the lack of relativistic conformal symmetry for z = 1 [sic], we are unable to directly apply the replica trick in the calculation of the EE. Thus, we will either resort to numerical methods, or to very special subsystems in which we can compute the EE analytically."

However, the references [16] describes how the replica trick may be applied in any free field theory. Conformal symmetry makes it easier to compute the entanglement entropy, but not impossible.

3-In the "Note added", it would be helpful to refer to an equation number or figure where their results may be seen to conflict.

4-In equation (2.3), $\phi$ should be defined.

5-After Figure One, there is insufficient discussion of the behavior. It is clear that the entanglement increases with $z$, and some arguments are given to say this is reasonable, but the authors miss the opportunity to make a broader point about why this must be or how general this effect should be. It might be good to have a prediction for how the dynamical exponents affects the entanglement and use numerics to confirm it. It also appears there might be a crossover behavior in $S(z)$ at $z \sim N_A = 20$. Above $z = 20$, each site directly interacts with sites outside the entanglement cut, so it is reasonable to expect the entanglement might scale differently in this regime.

6-Similarly after Figure 3, there is insufficient insight into the behavior of the numerics. One should be able to say something about the scaling of the EE.

7-The sentence in the conclusion "While the focus of our paper..." might be better suited for the abstract as it provides a good motivation for studying this topic.

8- Figures: the figures are bitmap images and thus do not render well when printed. The legends in particular become too pixelated to read. Using vector images (e.g. pdf, eps) would improve this. It may be advisable to make the figures colorblind-safe by using different symbols for data, in addition to different colors, when there are multiple sets of data on the same plot, such as in Figure 1.

9-Typos, grammar, notation (non-exhaustive). In the abstract: "entanglement entropies (EEs)"-> "entanglement entropy (EE)". Note added: "Dirichlet boundary condition" --- should be "conditions" In Equation (2.4) and below, it may be easier to read to re-define $J^{-1} \to J$. Figure 3: "as a function of the size of the subinterval". It may be clearer to say "as a function of the size $N_A$ of the subinterval". Figure 7: the caption refers to the parameters $\widetilde{J}_0/J_0 = 2^{20}$ but also $\widetilde{J}_0 = 0, 2^{-30},$ and $2^{-40}$. However, the legend refers to $\widetilde{J}_0 = 2^{-20}$ instead. Also, if $\widetilde{J}_0 = 0$, then $\widetilde{J}_0/J_0 = 0 \neq 2^{20}$. Please clarify the parameters.

---

## Round 2 · Referee Report · Anonymous (Referee 2) · 2017-7-4

Strengths

1) First results in an unexplored domain 2) Well written text 3) Combination of numerical and analytical findings

Weaknesses

1) Less clear results (conjectured or numerically suggested) about the EE in the more important linear segment sublattice case 2) No comments about other entropy measures (Rényi entropies, dynamical entropy) 3) Low quality figures

Report

In this paper the entanglement entropy (EE) of (1+1)-dimensional free Lifshitz scalar field theories is computed for different values of the dynamical exponent, z. The authors have considered two geometries for the subsystem: i) the traditional choice of a line segment of consecutive points, and ii) periodic sublattices. In the first case numerical calculations are performed and the qualitative features of the EE as a function of "z", and the length of the subsystem are analysed. Unfortunately no conjectures about the possible functional dependence are obtained through these studies. For periodic sublattices, which are somehow less interesting, there are analytical results, too. They have also studied the behaviour of the EE, when the Lifshitz theory is perturbed with relevant operators. The obtained results are physically motivated and can be considered as plausible. The paper is generally well written, the introduction gives the scientific background, mathematical derivation of the results are well documented. The literature is adequately cited.

Requested changes

1) Phi is not defined in (2.3). 2)The figures are of low quality, they should be replaced. 3) The Conclusions should be extended with some comments. What is expected to happen with other entropy measures, such as the Rényi entropy. What about the behaviour of the dynamical entropy after a sudden change of the parameters (global quench). 4) There are a few typos, such as "prove prove" in p. 15.

---

## Round 3 · Referee Report · Anonymous (Referee 1) · 2017-9-20

Strengths

1-Studies entanglement entropy in a new class of 1+1d theories.
2-Relatively clean presentation, writing and organization

Weaknesses

1-Numerical evidence in support of some claims is perfunctory.

Report

The paper is substantially improved by the changes from the previous version. The use of analytic arguments now provides a qualitative understanding of the effect of the dynamical exponent on entanglement, which is borne out in several of the numerical calculations performed. The main new result, Equation (2.11), is an interesting an straightforward generalization of $z=1$ results, which can be numerically confirmed to very high precision. It is unfortunate that Figure 1 shows somewhat large deviations from the predicted behavior, even at $z=1$ which should be well-understood. It may be appropriate to comment more fully on the non-universal behavior present in the numerical data or, better yet, reduce it (e.g. by going to smaller mass values or larger $z$).

In conclusion, many of the largest problems have been corrected and the paper now provides adequate novelty and understanding of its subject matter.

Requested changes

1 - The quality of the figures is still poor. In particular, Figure 2, 3, and 7 are still bitmapped. They should either be replaced by vector graphics, or raster graphics with a high enough resolution to print well. (PDF is a flexible format that can include both raster and vector images. Modern plotting programs such as matplotlib, mathematica, etc have options for exporting vector graphics. If raster graphics must be employed, one should use a resolution of at least 600DPI so that they print well.)

---

## Round 3 · Author Response

Dear editor,

We are grateful for the careful reading of the manuscript, and for the detailed report on it. We have significantly modified the version of the manuscript, and now in particular have an analytic understanding of subinterval EE in massless Lifshitz scalar theories in 1+1 dimensions using cMERA techniques. We have thus reorder section 2, with the new analytical results, which basically describe the scaling of subinterval EE with the dynamical exponent in a very simple way. We compare such analytical results with the previous numerics. In addition, we have also addressed nearly all the comments the referees brought up, and thank the referees for their criticism, which in our opinion made this paper much more complete. The only comments we did not address are the following:

1-One referee suggested comments on Renyi entropy/global quenches in our conclusion. We have not studied those aspects and hence feel we cannot say anything too meaningful in that regard with confidence.

2-One referee suggested us to move a sentence from the conclusion to the introduction/abstract. We believe the paper is a self-contained research within field theories with Lifshitz symmetry. Therefore, we relegated comments on possible applications to other areas and further developments to the conclusion.

3-We are not sure about the reason of the poor resolution of the figures. The referees suggested using pdf format instead of
bitmap format, but the format we are using is indeed pdf. Hopefully this wouldn't be an issue again.

We hope that the present version convinces the editor to accept the manuscript for publication.

With best regards,

The authors

---

## Round 3 · List of Changes

1- Fixed typo in v2 equation (1.2) and changed the paragraph around it into a footnote.

2- Clarified what we mean by "Due to the lack of relativistic conformal symmetry for z = 1 [sic], we are unable to directly apply the replica trick in the calculation of the EE. Thus, we will either resort to numerical methods, or to very special subsystems in which we can compute the EE analytically."

3- Added relevant equation number in "Note added."

4-In v2 equation (2.3), defined phi.

5-Obtained analytic results for subinterval EE in massless Lifshitz theories using cMERA techniques. Section 2 is now drastically extended to include these analytic results and how they compare with the numerics.

6-We commented on the observed crossover behavior in Fig 3 around $z~N_A$.

7-Corrected typos.

---

## Editorial Decision

published